# A study on assessing the toxic effects of ethyl paraben on rohu (*Labeo rohita*) using different biomarkers; hemato-biochemical assays, histology, oxidant and antioxidant activity and genotoxicity

**Hasnain Akmal**[1], **Shabbir Ahmad**[1], **Muddasir Hassan Abbasi**[1], **Farhat Jabeen**[2], **Khurram Shahzad**[ID][1]*

1 Department of Zoology, University of Okara, Okara, Punjab, Pakistan, 2 Government College University, Faisalabad, Punjab, Pakistan

* dr.khurram@uo.edu.pk

**Data Availability Statement:** All relevant data are within the manuscript and its Supporting information files.

## Abstract

Parabens are being used as preservatives due to their antifungal and antimicrobial effects. They are emerging as aquatic pollutants due to their excessive use in many products. The purpose of this study was to determine the toxic effect of ethyl paraben ($C_9H_{10}O_3$) on the hematobiochemical, histological, oxidative, and anti-oxidant enzymatic and non-enzymatic activity; the study also evaluates the potential of ethyl paraben to cause genotoxicity in Rohu *Labeo rohita*. A number of 15 fish with an average weight of 35.45±1.34g were placed in each group and exposed to ethyl paraben for 21 days. Three different concentrations of ethyl paraben, i.e., $T_1$ (2000μg/L), $T_2$ (4000 μg/L), and $T_3$ (6000 μg/L) on which fish were exposed as compared to the control $T_0$ (0.00 μg/L). Blood was used for hematobiochemical and comet assay. Gills, kidneys, and liver were removed for histological alterations. The results showed a significant rise in all hemato-biochemical parameters such as RBCs, WBCs, PLT count, blood sugar, albumin, globulin, and cholesterol. An increase in aspartate aminotransferase (AST) and alanine transaminase (ALT) levels directed the hepatocytic damage. Histological alterations in the liver, gills and kidneys of fish were found. Ethylparaben induces oxidative stress by suppressing antioxidant enzyme activity such as SOD, GSH, CAT and POD. Based on the comet assay, DNA damage was also observed in blood cells, resulting in genotoxicity. Findings from the present study indicate that ethyl paraben induces hemato-biochemical alterations, tissue damage, oxidative stress, and genotoxicity.

## Introduction

Aquatic toxicity is a serious health hazard for aquatic ecosystems, including primary producers, microorganisms, fish, and humans, because water bodies contain fertilizers, insecticides, pesticides, and pharmaceutical chemicals that are added via leaching, runoff, and spray drift [1]. Toxic chemicals discharged from anthropogenic activities [2] become part of the

**Funding:** The authors received no specific funding for this work.

**Competing interests:** The authors have declared that no competing interests exist.

aquaculture food chain and cause serious health problems for humans who feed on these contaminated products [3].

To meet the requirements of an increasing population, the use of disinfectants, drugs, preservatives, and antifoulant (biocides) chemicals to remove bacteria and fungi has increased in industries. Biocides in aquatic environments have toxic effects with demonstrated detrimental impacts [3]. Parabens are produced when p-hydroxybenzoic acid is esterified in a catalyst and alcohol [4]. They are biocides and derivatives of esters of p-hydroxybenzoic acid [5]. The antimicrobial and preservative abilities of parabens were first reported in 1924 [6]. For more than 70 years in food, pharmaceutical, and cosmetic industries, it has been widely used as a preservative.

Chemical inertness, low cost, worldwide acceptance, and reduced toxicity are the features of paraben that make it the favorite compound of chemists compared to all other biocides used in food, pharmaceutical, and cosmetic products. Parabens are tasteless, odourless and effectively neutral in PH. They do not harden or discolor cosmetic products [7]. The most common members of the paraben family are methylparaben, ethylparaben, propylparaben, butylparaben, and benzyl paraben. They are used separately and in combination with others. Mainly in the aqueous phase, influent contain (>97%) and wastewater treatment plants contain 79,600 ng/L concentrations of parabens [8].

Estrogenic effect of parabens on fish was observed by Bjerregaard, Andersen [9]. However, the impact of parabens on fish tissues and hematology has not been evaluated.

Parabens enter the human body via dermal exposure, exhalation, and ingestion [10]. Their negative effects on the body cause diabetes, neurotoxicity, adult obesity, and a reduction in the quality and quantity of human sperm. Parabens have been found in the tumour cells of breast cancer, which indicates that they increase the chances of cancer [11]. These toxic compounds harm endocrine function by interfering with endocrine system [12].

Ethylparaben ($C_9H_{10}O_3$) is a member of the paraben family with a short chain of alkyl groups. The cell penetrability, stability, and antibacterial activity of parabens increase with the increase in the chain [13]. The increase in the basal and vitellogenin levels in the blood by ethylparaben shows oestrogenic effects [4]. Frederiksen, Taxvig [14] studies on the pregnant rat observed that ethylparaben continues its exposure to the fetus by accumulating in the maternal plasma and placenta of the fetus.

The present study aims to evaluate ethyl paraben's toxic effects on freshwater fish, *Labeo rohita*, as the concentration of ethyl paraben in aquatic environments increases day by day, which is harmful to aquatic organisms. Fish is a source of protein and part of the food chain; its consumption after being caught from polluted water resources is harmful to humans.

## Materials and methods

### Laboratory setting and animal handling

All the experiments were performed in the laboratories of the Department of Zoology and Department of Fisheries, University of Okara, Okara, Pakistan. The standard protocols were followed for the preparation of chemicals and reagents. The ethical research committee (ERC) of the University of Okara, Pakistan, provided ethical approval (UO/ERC/2023/36) for fish handling. The fish were caught from the ponds at Head Balloki, District Kasur, Pakistan and then transported to the laboratories. During the fish's transport, the standard FAO protocol was followed; thereby no mortality occurred. Before transferring these fish to the experimental tank, they were first acclimatized with 0.1% $KMNO_4$ solution for 7 days to prevent fungal and microbial attacks. Commercially prepared feed (20% protein) was given to all the fish. Dissolved oxygen (DO) and pH of the water in the aquarium ranged from 6.25±2.01 ppm and 7.2

±1.02, respectively. The ammonia and nitrate levels varied between 0.19–0.47 ppm and 0.04–0.14 ppm, respectively. The water temperature varied from 25.4 to 30.2˚C in the experimental tanks.

## Chemical

Ethylparaben ($C_9H_{10}O_3$) of 99.9% purity was purchased from MACLIN, China. A standard protocol was followed to make all the reagents of analytic grade.

## Experimental design and treatment

Four groups of fish were formed for experimental trial: a control group ($T_0$) and three treatment groups ($T_1$, $T_2$, and $T_3$) each with a triplicate. 15 fish were placed in each group (n = 15). A stock solution of ethylparaben was prepared using analytic-grade ethanol. $LC_{50}$ value of *ethyl* paraben for 48 hours was determined (LC50 = 12000 µg/L) by using the probit analysis method [15]. Fish groups $T_0$, $T_1$, $T_2$, and $T_3$ were exposed to different concentrations of ethylparaben 0 µg/L, 2000µg/L, 4000µg/L and 6000µg/L respectively, for 21 days. 90% of the water in each glass aquarium was replaced with fresh water every second day.

## Haematological and biochemical analysis

The blood was procured by direct heart puncture and from the abdominal vein using disposable syringe and 22-gauge needle (Molnar 1960) [16] (n = 3). The blood was collected in an EDTA tube for hematology and was transpired into mottled red top tubes for biochemical analysis. The haematological parameters, including RBC, hematocrit, haemoglobin, MCH, MCV, and MCHC, were assessed by an automatic haematological analyser (HEMA D6031). The white blood cell count was determined by following Ghaffar, Hussain [17].

For biochemical assay, the blood was centrifuged at 4˚C and 1,000 rpm for 10 min to separate plasma. Total protein, albumin, globulin, AST, ALT, total cholesterol, HDL-C and LDL-C were analysed using a chemistry analyzer (AU480, Beckman coulter, US). According to Waterborg [18] method, the protein content of the samples was calculated using bovine serum albumin as standard. Glucose level was determined using the O-Toluidine assay [19].

## Histology

The protocol of Barse, Chakrabarti [20] was used for histological determination. At the end of the trial, fish were anesthesized with clove oil (50 µl/L) and dissected. Internal organs such as the liver, kidney, and gills were removed from the fish (n = 3). Samples were stored in 90% alcohol until further processing. They were first fixed in neutral buffered formalin, and then in 70% alcohol in four steps. The tissue was embedded in paraffin after preservation and dehydrated using a progressive alcohol series. 4µm sections were stained with hematoxylin and eosin and viewed using an Olympus BX40 microscope. The photographs were taken with a Nikon (SC 35 type 12) camera with a shutter speed of 30 seconds.

## Biochemical assay

**ROS assay.** The activity of reactive oxygen species (ROS) was observed by following Tvrda [21] with a little modification. Homogenised mixture of tissue was mixed with phosphate buffer (0.1 M, pH 7.4). The cells were solubilized using 2M potassium hydroxide (KOH), centrifuged and determined the absorbance in supernatant between 530 nm and 630 nm.

**TBARS.**   The standard protocol of Yagi and medicine [22] was used to measure the activity of TBARS. The homogenised tissue sample was treated with 0.1% Triton X100. 20 μL,40 μL, 60 μL, and 80 μL solution mixtures were mixed with thiobarbituric acid and then heated at 95˚C, centrifuged and determined the absorbance in supernatant at 532nm. Tetrametoxy propane was used as standard.

**Catalase.**   The protocol of Sinha [23] was used to analyse catalase. The reagents were Phosphate buffer (0.1 M, pH 7.0) and 30% $H_2O_2$ solution. The tissue was homogenised in phosphate buffer. The reaction mixture includes1.95 mL phosphate buffer, 0.1 mL of $H_2O_2$, and 0.05 mL tissue extract. Change in absorbance was recorded at 240 nm.

**Superoxide dismutase.**   The protocol was according to Beauchamp and Fridovich [24]. For the analysis of superoxide dismutase the reagents were methionine (150mM), riboflavin (24μM), phosphate buffer (0.1M, pH 7.5), nitro blue tetrazolium (840 μM), and $Na_2EDTA$ (1.2 mM). The reaction mixture includes 0.05mL tissue extract, 0.25 mL NBT, 0.25 mL. Na2EDTA, 0.25 mL methionine, 1.95 mL phosphate buffer, and 0.25 mL riboflavin. The extract was pipetted in 4 glass tubes. Four more glass tubes were prepared, adding 0.05 mL of phosphate buffer instead of enzyme extract. Three tubes from each set were then placed on shaker at 25˚C in fluorescent light for 15 minutes and the last one was kept in dark at 25˚C (reference sample). After the incubation period, the change in the absorbance was measured at 560 nm using respected dark-incubated sample as reference for test samples for each series. The SOD activity was expressed in terms of relative enzyme activity (U/mg protein).

**Glutathione.**   The activity of glutathione (GSH) was evaluated using Carlberg and Mannervik [25]. The activity was observed by measuring NADPH at 340 nm. The reaction mixture includes 16 M oxidised glutathione, 8 mM NADPH, 50 mM $MgCl_2$, and phosphate buffer (0.1 M, pH 7.5). The reaction is initiated by adding supernatant to the reaction mixture.

**Peroxidase.**   Activity of peroxidase in homogenized mixture was measured by formation of TBARS and quantified as malondialdehyde (MDA) equivalents by following Buege and Aust [26].

## Comet assay

The comet assay protocol of [27, 28] has been followed. Slides were examined under a fluorescence microscope at 400X. Comet IV Software, was used for scoring the comet [29].

## Statistical analysis

Statistical analysis was done by applying ANOVA to IBM SPSS (Version 21) software at $p < 0.05$ level of significance. Graphic Prism Version 9.3.1 was used for graphical representation.

## Results

### Hematology

The results of hematological analysis of *L.rohita* after exposure to ethyl paraben are shown in (Fig 1). After exposure to a high dose (6000μg/L) of ethylparaben, the concentrations of HGB, WBC, RBCs, HCT, MCV, MCH, MCHC and platelets count increased significantly. When they were exposed to medium dose (4000 μg/L), some of these parameters such as HGB, WBC, MCH, MCHC and MCV showed significant elevation but others like RBCs, HCT, and PLT did not show significant increase. Hematological parameters like, HGB, RBCs, HCT MCH MCHC and PLT after low dose (2000 μg/L) treatment did not show significant increase but in the case of WBCs and MCV this increase was significant.

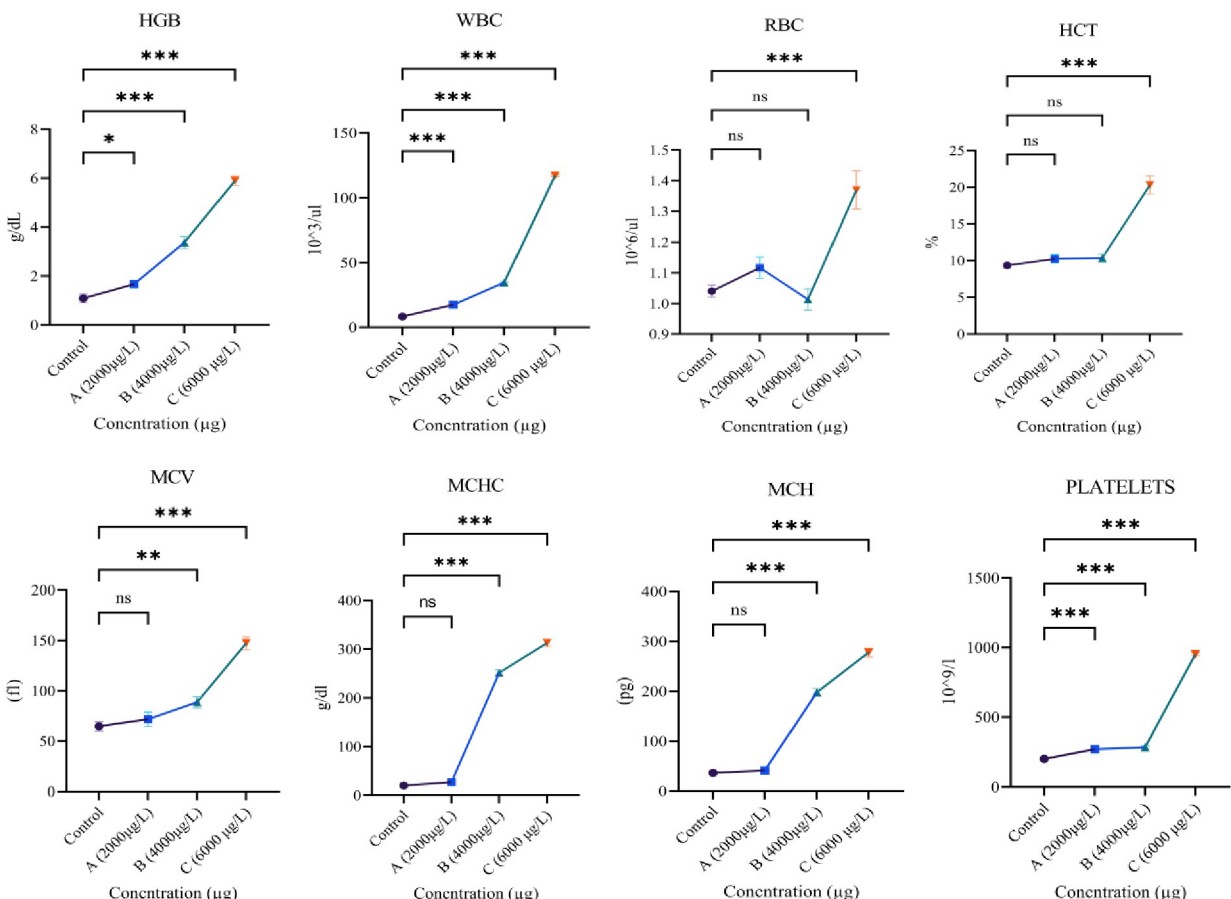

**Fig 1. Showing variations in blood parameters of *L.rohita* treated with different concentrations of ethylparaben.** All values are mean ± SD at p<0.05 level of significance. Steric (*) shows significant steric (**) shows highly significant.

## Biochemical parameters

The lipid profile of *L.rohita* after being treated with ethylparaben is shown in (Fig 2). After exposure to high (6000μg/L) and medium (4000μg/L) doses of ethylparaben, the lipid profile of *L.rohita*, such as cholesterol, triglycerides, HDL cholesterol, LDL cholesterol, and VLDL cholesterol, showed highly significant increase. Low dose (2000 μg/L) treatment of *L.rohita* showed that some parameters like cholesterol, LDL cholesterol and HDL cholesterol neither increased nor decreased significantly. VLDL cholesterol showed a significant increase at low doses and triglycerides showed a highly significantly increase.

Results of blood sugar, total protein, albumin, globulin, liver functioning enzymes alanine transaminase (ALT), and aspartate aminotransferase (AST) are shown in Fig 3. All these parameters showed significant increases when exposed to different concentrations of ethyl paraben as compared with control group.

## Oxidative stress and antioxidant response

Fish tissues, liver, kidney and gills were used to evaluate oxidative stress ROS and TBARS (Table 1). The responses of antioxidant enzymes GSH, SOD, CAT, and POD are shown in (Fig 4). The measurements of ROS and TBARS from the liver, gills, and kidney showed a significant

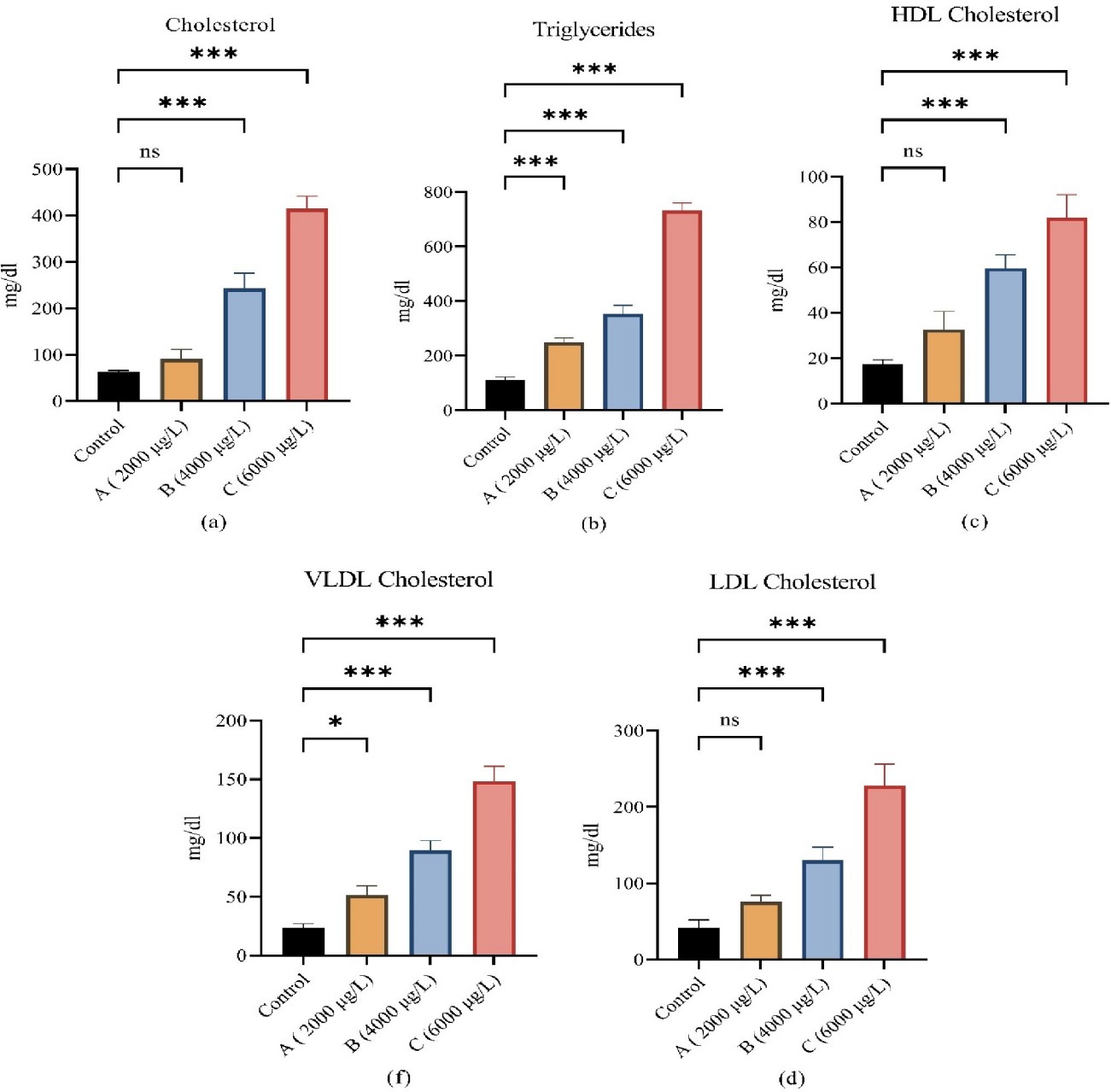

**Fig 2.** (a) showing alterations in blood cholesterol, (b) showing alterations in triglycerides, (c) showing alterations in HDL cholesterol, (d) showing alteration in LDL cholesterol, and (e) showing alterations in VLDL cholesterol between control and all three treated groups (low, medium and high doses). All values are mean ± SD at p<0.05 level of significance. Steric (*) shows significant steric (**) shows highly significant.

increase in these biomarkers after treating *L. rohita* for 21 days with *ethyl*-paraben for 2000μg/L, 4000 μg/L, and 6000 μg/L as compared with the control group. The measurements of all anti-oxidant biomarkers, i.e., GSH, SOD, CAT, and POD, showed significant reduced values ($p \leq 0:05$) in the liver, gills, and kidney.

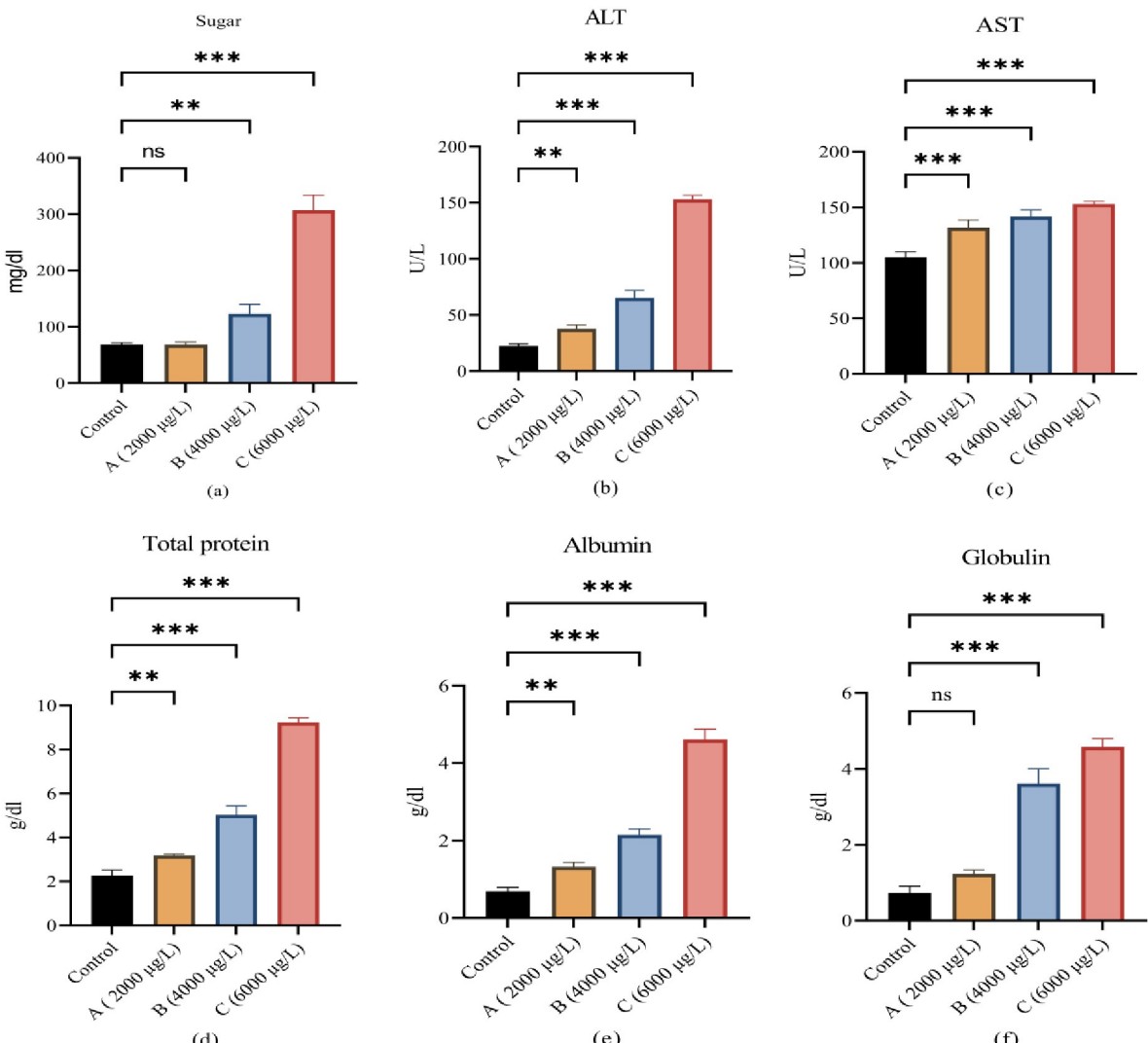

**Fig 3.** (a) showing alterations in total protein, (b) showing alterations in globulin, (c) showing alterations in albumin (d) showing alteration in aspartate aminotransferase, (e) showing alterations in alanine transaminase, and (f) showing alterations in blood glucose level between control and all three treated groups (low, medium and high doses). All values are mean ± SD at *p<0.05* level of significance. Steric (*) shows significant steric (**) shows highly significant.

## Histology

Histology of the gills, liver and kidneys of *L. rohita* is shown in Figs 5–7. Normal morphological structure i.e. secondary lamellae coated with simple squamous epithelial cells was observed during gill histology of control group (5a). When these fish were exposed to different concentrations of ethylparaben, alterations in histology of gills were observed in treated group (5b-5d) such as lamellar disorganization, fusion of primary lamellae, lamellar aneurysm, hyperplasia of epithelial cells, and bone cell deformities. The normal arrangement of glomeruli in the kidney of *L. rohita* was observed during the histology of the control group (6a). Histology of the treated group (6b-6d) showed glomerular expansion and absence of Bowman's capsule, tubular cells with hypertrophied nuclei, Sinusoidal spaces, cluster nuclei formation, Melanomacrophages, and damaged parenchyma cells.

**Table 1. Oxidative stress in the liver, gills and kidney of *L.rohita* exposed to different concentrations of ethylparaben.**

| Liver | T0 (Control) | T1 (2000 µg/L) | T2 (4000 µg/L) | T3 (6000 µg/L) |
|---|---|---|---|---|
| ROS (optical density) | 0.3500±0.03606 | 0.4533±0.0802* | 0.830±0.040* | 1.02±0.075** |
| TBARS (nmol/TBARS formed/mg protein/ min) | 36.87±0.075 | 43.65±0.060* | 53.13±0.080** | 60.12±0.105** |
| **Gills** | | | | |
| ROS (optical density) | 0.32±0.085 | 0.44±0.07* | 0.54±0.08** | 0.61±0.08** |
| TBARS (nmol/TBARS formed/mg protein/ min) | 39.89±0.085 | 46.76±0.075* | 63.65±0.075** | 67.34±0.085** |
| **Kidney** | | | | |
| ROS (optical density) | 0.60±0.085 | 0.64±0.07 | 0.72±0.08* | 0.81±0.095* |
| TBARS (nmol/TBARS formed/mg protein/ min) | 28.65±0.07 | 35.12±0.095* | 39.62±0.09* | 41.39±1.23** |

ROS = Reactive oxygen species; TBARS = Thiobarbituric acid reactive substance

Steric (*) showing significant steric (**) showing highly significant.

Liver histology of the control group (7a) showed no change in the liver of *L.rohita*. The hepatic parenchyma showed sinusoid-limited hepatocyte plaques. The hepatocytes have polygonal shapes, granular cytoplasm, and rounded nuclei. They were arranged in cord. After exposure to ethylparaben, alterations in the hepatocytes of *L.rohita* were observed (7b-7d). Histology of the treated groups showed pyknotic nuclei, eosinophilia granules, sinusoidal spaces, necrosis, and melano-macrophages in liver tissues.

## Genotoxicity

The blood cells of *L.rohita* were used for alkaline comet assay to evaluate the potential of ethyl paraben to cause genotoxicity in *L.rohita* (Fig 8). The results of alkaline comet assay indicate that ethylparaben causes DNA damage in blood cells. This damage increases with the increase in concentrations of ethyl paraben. The percentage of tail DNA observed at 15.69±0.74 after exposure to the high dose as compared with the control group at 0.91±0.55 reveals a significant difference. By increasing concentrations of ethyl paraben in A (0µg/L), B (2000µg/L), C (4000µg/L), and D (6000µg/L), the difference increases significantly. Olive tail moment (OTM) shows substantial variations as well; at the high dose (6000µg/L), it was 8.37±2.45, whereas at the control (0µg/L), it was 0.63±0.11. Tail DNA and olive tail moment (OTM) increase with the increase in the concentration of ethylparaben.

## Discussion

Aquatic toxicity has become a major concern due to the high proportion of toxicants being released into aquatic ecosystems that have gone unchecked in the recent years. The majority of these toxicants are carcinogenic and cause anthropogenic effects on humans as they feed on aquatic organisms and plants [30]. Parabens belonging to one of these aquatic toxicants have anthropogenic effects and cause regional and localized pollution problems around the world [31, 32]. These toxicants, such as paraben, cause environmental disasters [33].

Hematological parameters are considered indicators for detecting toxicology. Different toxicants like parabens cause alterations in hematological parameters [34]. These alterations have been considered in response to high oxygen demands and gas transfer during stress conditions. The spleen and liver reactivate erythropoiesis in hypoxic conditions to compensate for increased oxygen demand in peripheral tissue [35]. In the present study, the elevation in the concentrations of HGB, WBC, RBCs, HCT, MCV, MCH, MCHC, and platelet count has been observed after exposure to different concentrations of ethyl paraben to cope with stress

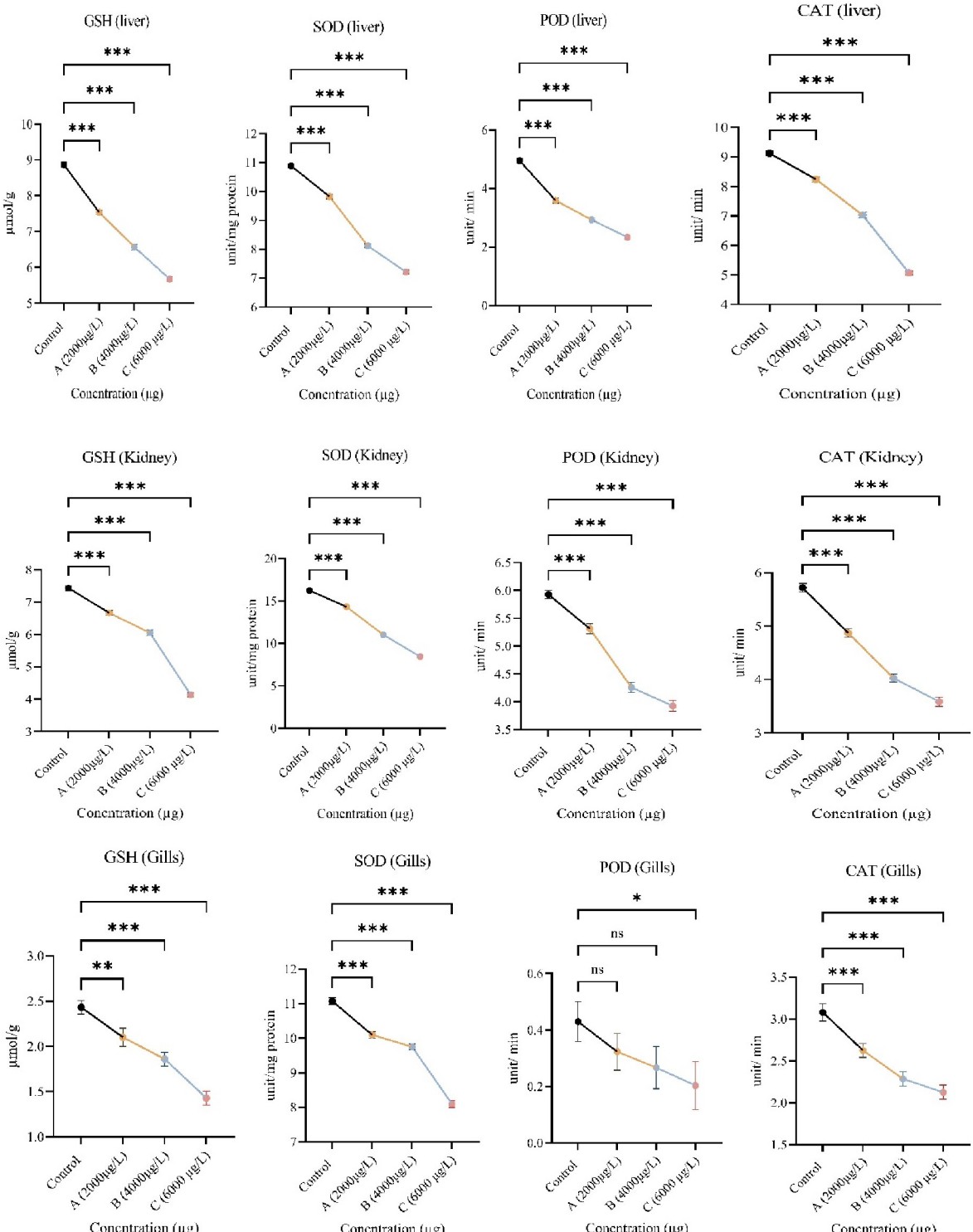

**Fig 4. Showing decrease in level of antioxidant enzymes in different tissues (liver, gills and kidney) of *Labeo rohita* after exposure to higher concentrations of ethyl paraben.**

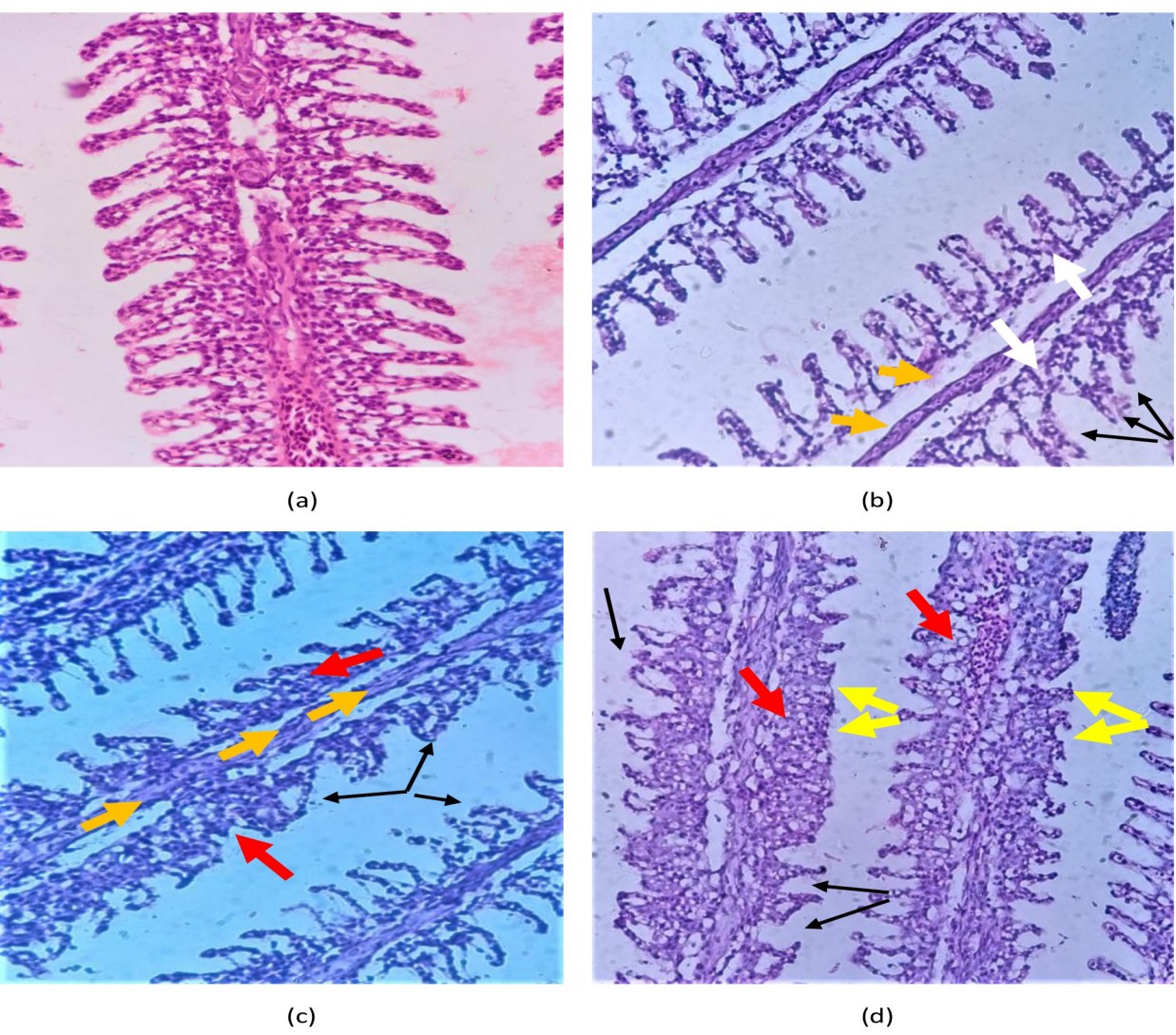

**Fig 5. Shows the gills histopathology of *L. rohita* after exposure to ethyl paraben for 21 days (40X).** The images include (a) control group, (b-d) 2000 µg/L, 4000 µg/L, and 6000 µg/L. The gills of the control group showed normal histopathology, whereas the other treated groups showed lamellar disorganization (black arrow), fusion of primary lamellae (red arrow), lamellar aneurysm (yellow arrow), hyperplasia of epithelial cells (white arrow), and bone cell deformities (orange arrow).

conditions. Our results are supported by the study done by Park, Lee [36], which has observed an increased level of RBCs, HGB, Hct, MCV, MCH, and MCHC in rats after exposure to parabens. Similar rises in RBCs, HGB, and Hct in male rates have been observed by İNKAYA, KARABULUT [37]. The higher concentration of WBCs and PLT count after exposure to ethyl paraben has been observed in the present study. Therefore, a similar decrease in the concentration of WBCs and PLT count after exposure has been detected by İNKAYA, KARABULUT [37]. No previous work has been found on ethylparaben exposure to fish. This increase in WBCs is due to the infection and inflammation caused by ethylparaben exposure to the body. During the infection state, the immune system activates, and white blood cell concentration increases [38]. Ethylparaben exposure causes inflection due to which platelet count increases from low to high concentration as compared to the control. This increase in platelet count is

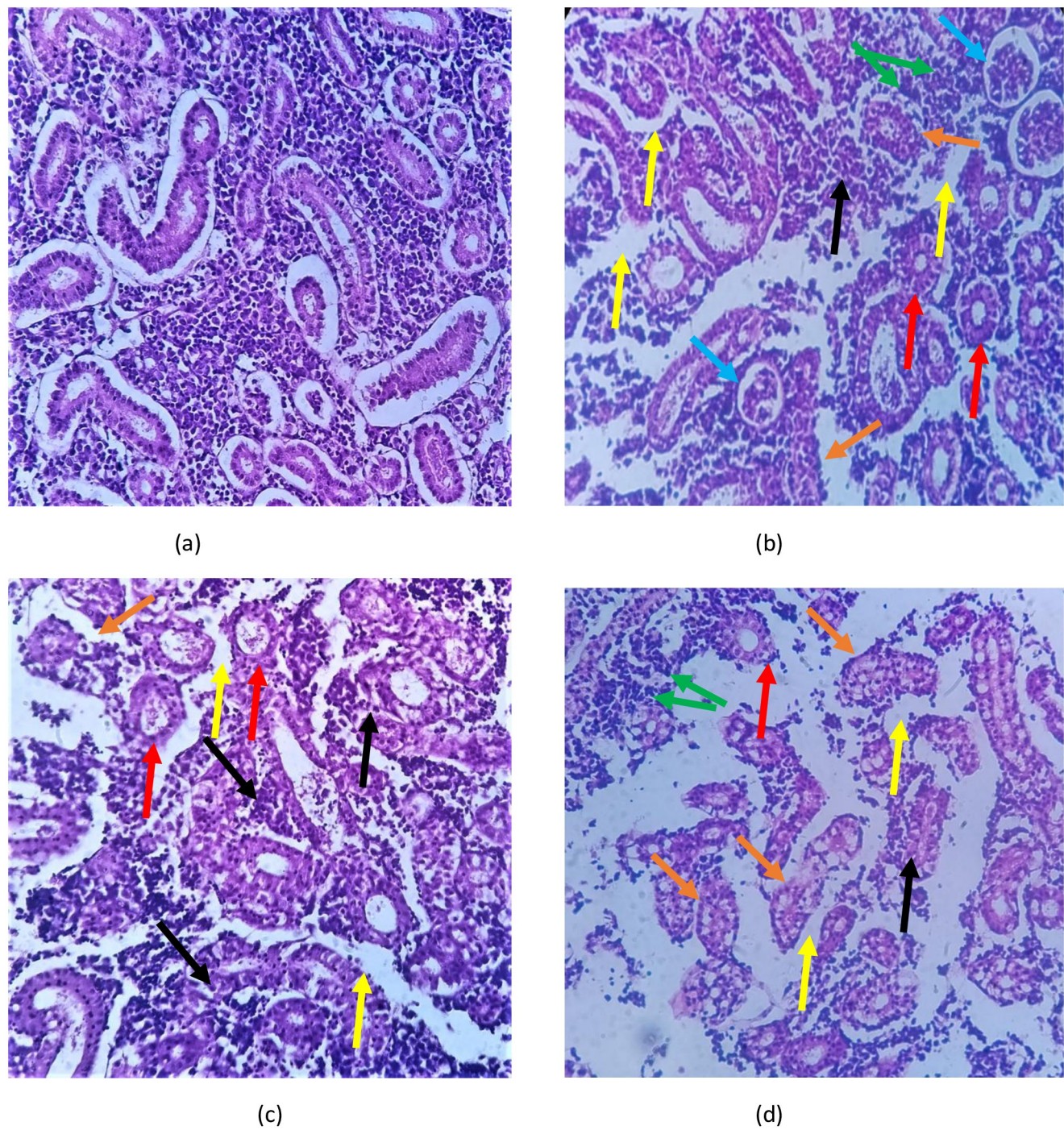

(a)

(b)

(c)

(d)

**Fig 6. Shows the kidney histopathology of *L. rohita* after exposure to ethyl paraben for 21 days (40X).** The images include (a) the control group, (b-d) 2000 μg/L, 4000 μg/L, and 6000 μg/L. The kidney of the control group showed normal histopathology, whereas the other treated groups showed glomerular expansion and absence of Bowman's capsule (black arrow), tubular cells with hypertrophied nucleus (red arrow), Sinusoidal spaces (yellow arrow), cluster nuclei formation (green arrow), Melano-macrophages (blue arrow), and damaged parenchyma cells (orange arrow).

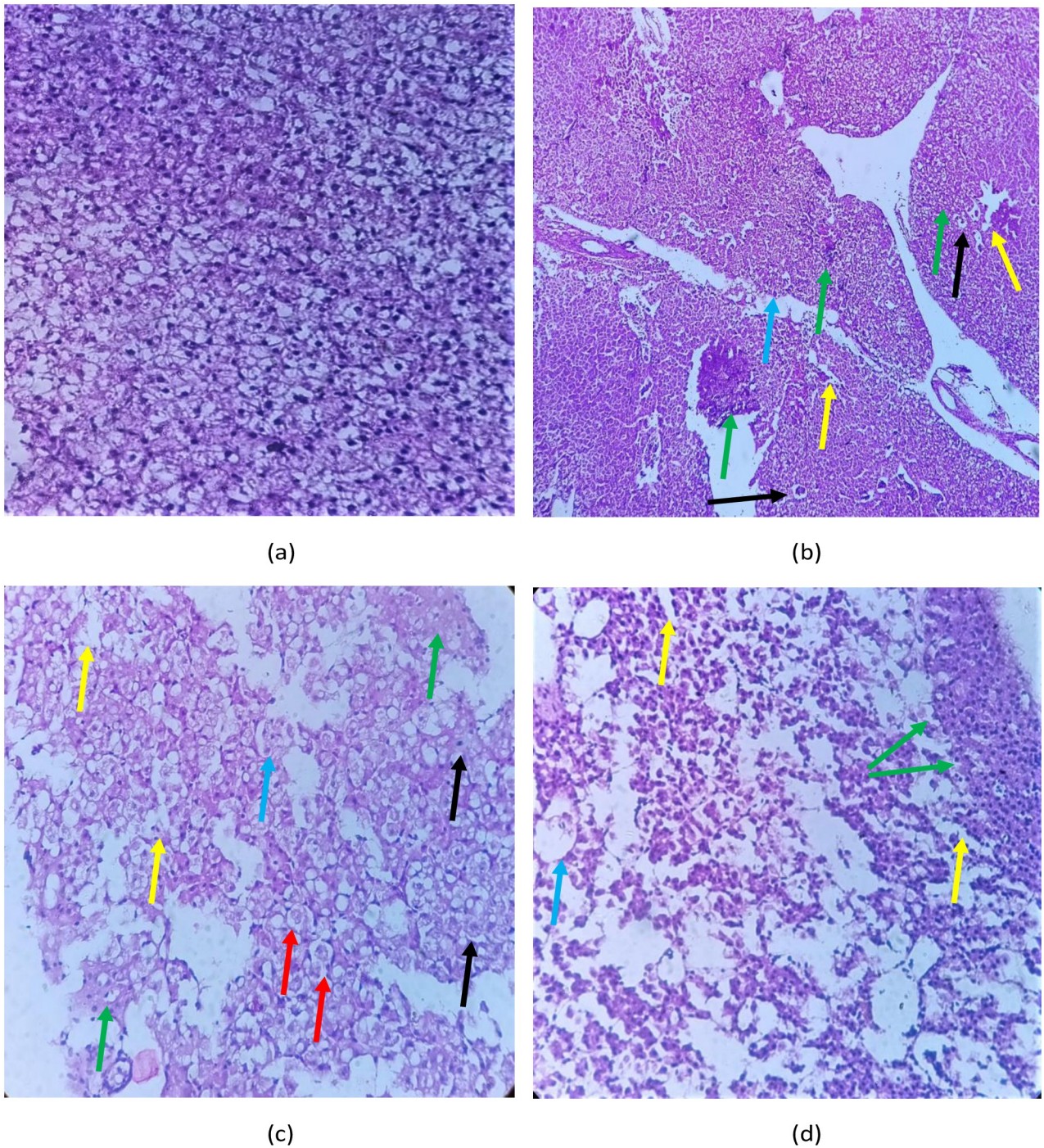

**Fig 7. Shows the liver histopathology of *L. rohita* after exposure to ethyl paraben for 21 days (40X).** The images include (a) the control group, (b-d) 2000 μg/L, 4000 μg/L, and 6000 μg/L. The liver of the control group showed normal histopathology, whereas the other treated groups showed Pyknotic nuclei (black arrow), eosinophilia granules (red arrow), Sinusoidal spaces (yellow arrow), necrosis (green arrow), and Melano-macrophages (blue arrow).

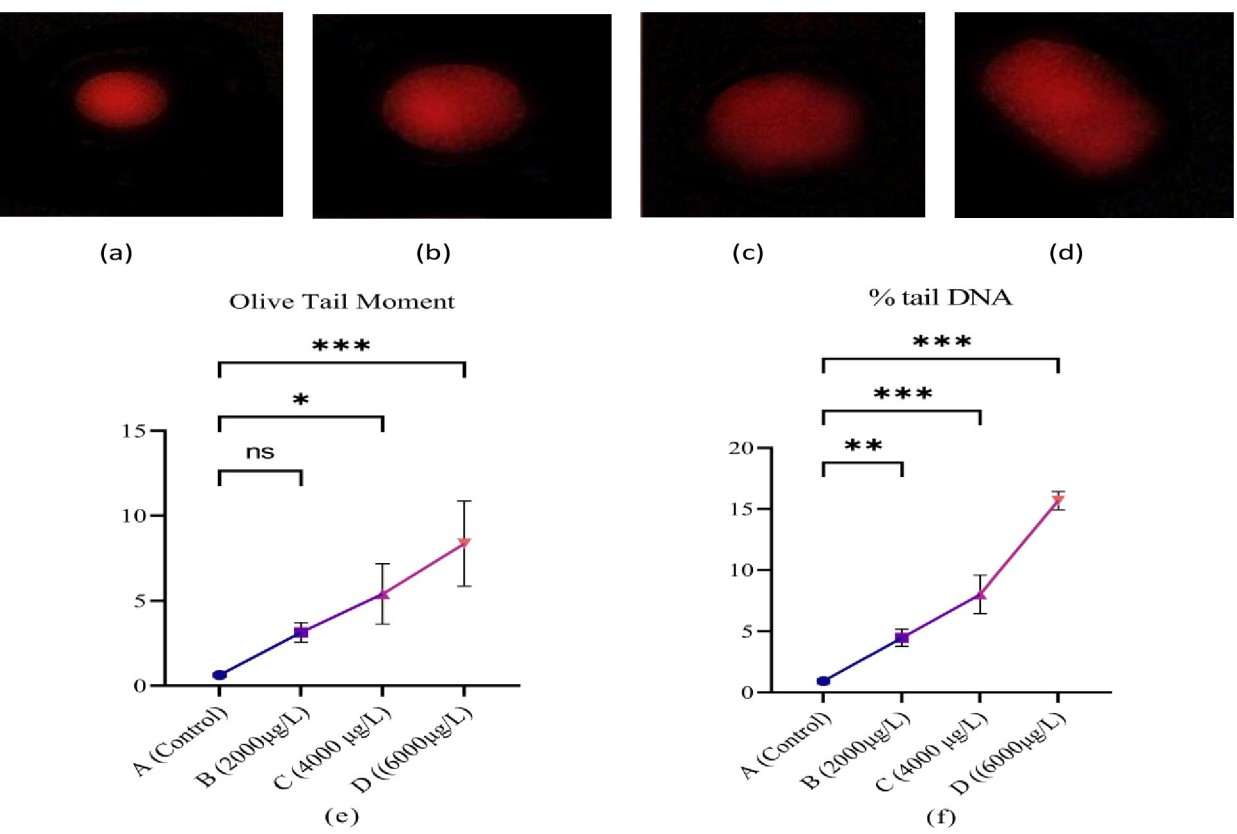

**Fig 8.** Genotoxicity of Ethyl paraben in RBCs. Comet assay: (a) reference control olive and (b-d) reference treated comet. All values are mean ± SD with p<0.05 level of significance.

due to thrombocytosis—a condition that causes an increase in platelet concentration in the infectious state [39].

Parabens such as ethylparaben, being endocrine disruptors, interfere with normal function in many processes and alter them like the immune system, lipid homeostasis, glucose level, and thyroid functioning [40]. Many hormones, like testosterone and oestrogen, are prepared from lipids. Results of the lipid profile after being exposed to different concentrations of ethyl paraben showed a significant increase in cholesterol, HDL cholesterol, LDL cholesterol, and VLDL cholesterol as compared to the control group. Kim and Chevrier [41] also recorded the same increase in cholesterol, triglycerides, HDL cholesterol, LDL cholesterol, and VLDL cholesterol in humans, both male and female. Lipids cannot breakdown in the blood and must be delivered to and from cells via low-density lipoproteins (LDL) and high density lipoproteins (HDL). High density lipoprotein cholesterol (HDL-C) transports cholesterol from the arteries to the liver. As a result, excessive serum cholesterol levels may be caused by hepatic dysfunction [42]. Triglycerides act as a source of immediate and stored energy. Due to high energy demand in stress conditions, the fish showed high triglyceride levels. The current study evaluates the toxic effect of ethyl paraben on hepatic tissues. For liver alanine transaminase (ALT) and aspartate aminotransferase (AST), blood sugar, total protein, albumin, and globulin are good biomarkers for hepatic toxicity. The results of serum biochemical showed that all these parameters were elevated significantly as compared to the control group. These results were supported by the findings of Beriry, Atef [43] and Darbre and Harvey [44]. The increases in

liver enzymes occur directly in response to liver toxicity caused by ethylparaben in hepatocytes. The primary function of the liver is to detoxify toxic chemicals and produce antioxidant enzymes such as catalase that break down hydrogen peroxide. That's why the liver is a primary organ to be damaged due to paraben toxicity [45]. Blood glucose levels also increased after exposure to ethylparaben. Similar increases in blood sugar were recorded by Bellavia, Chiu [46] in pregnant women after they put them at risk for gestational diabetes. No previous study on lipid profile and blood glucose level was found on the fish exposed to ethylparaben in a laboratory or natural environment.

Biochemical alterations are confirmed by histological changes in the current study. In fish, gills are primary organs in contact with water and are involved in osmoregulation, respiration, and excretion. Hence, they remain the primary target for toxicants. Histology of the gills revealed that after exposure to ethylparaben, the gills showed lamellar disorganization, fusion of primary lamellae, lamellar aneurysm, hyperplasia of epithelial cells, and bone cell deformities of secondary lamellae. The alterations in gills, like epithelial lifting, hyperplasia, and fusion of epithelial cells, were also reported by Camargo and Martinez [47]. A similar alteration in histology of the gills after exposure to paraben, as recorded by Beriry, Atef [43] supports our present study.

The kidney, being a vital organ in osmoregulation and excretion, also remains an easy target for toxicants [48]. The most common defects caused by water toxicants are tubule degeneration, dilation of capillaries, and degeneration of parenchyma cells of the kidney Irnidayanti, Fatona [49]. Similar changes came out as results of kidney histology of *L. rohita* like, glomerular expansion and absence of Bowman's capsule, tubular cells with hypertrophied nuclei, Sinusoidal spaces, cluster nuclei formation, Melano-macrophages, and damaged parenchyma cells as compared to the control, which showed a normal arrangement of the glomerulus in the kidney of *L. rohita*.

The primary organ in the body involved in detoxification and biotransformation of toxicants is liver. Therefore, it is one of the most affected organs due to aquatic toxicity [50]. A normal liver has regular shaped hepatocytes, cytoplasmic vacuolation, and a nucleus in a lateral position. The cytoplasm of hepatocytes contains lipids and glycogen inside vacuoles, which are used in normal metabolic functions. During stressful conditions, glycogen inside hepatocytes is depleted because glycogen acts as a reserve for glucose. Due to these large spaces (sinusoidal spaces), hepatocytes are formed [51]. The increase in vacuolization is an indication of metabolic damage due to the toxicity caused by ethyl paraben. The histology of the liver showed pyknotic nuclei, eosinophilia granules, Sinusoidal spaces, necrosis, and Melano-macrophages.

The results of ethylparaben exposure show that ethylparaben causes biochemical alteration and fluctuations in enzymatic activity. Oxidative stress is used to find out the damage caused by ROS (reactive oxygen species) and TBARS (Thiobarbituric acid reactive substances) such as hydroxyl radicals ($OH^-$), hydrogen peroxides ($H_2O_2$), and superoxide ($O_2^-$). A large amount of chemicals has toxic effects, such as inducing oxidative stress. The body's antioxidant defence systems include SOD, catalase, POD, and GSH. Xenobiotics such as ethylparaben suppress antioxidant activity, due to which large number of free radicles accumulate and damage cellular components like DNA, and carbohydrates [52]. Soft tissues of the body, including the liver, kidney and gills are more susceptible to free radicals' attacks because they are involved in metabolism, excretion etc.

The results of oxidative stress biomarkers such as TBARS and ROS showed that ethyl paraben exposure induces oxidative stress. In tissue damage, the primary biomarker for lipid peroxidation is TBARS assay. TBARS assay is the measurement of MDA (malondialdehyde) [53]. Due to being lipophilic in nature, ethylparaben can easily cross the plasma membrane and increase MDA level and consequently increase ROS. A high level of ROS suppresses the

antioxidant system, which works as an antagonistic to oxidative enzymes. Elevated ROS and decreased antioxidant systems change redox potential of plasma membrane and cause oxidative stress in soft tissues such as hepatocytes, renal tissues, and gill filaments. After exposure to ethylparaben, elevated levels of Lipid peroxidase (POD) were recorded in the tissues. Our results are supported by the findings of Shah and Verma [54] and Silva, Serrano [55].

A significantly decreased level of GSH (glutathione) was recorded in fish *L.rohita*. GSH is an antioxidant and free radical scavenger. Shah and Verma [54] also reported a similar decreased level of GSH after exposure to parabens. Another study by Silva, Serrano [55] shows that parabens exposure significantly increases GSH levels. This suppression of GSH level indicates the high concentration of free radicles. The number of free radicals is greater than the scavenging ability of GSH. The first line of defence against ROS-induced damage is enzymatic anti-oxidants, including catalase, SOD, and GSH [56, 57]. Our results indicate significant decrease in antioxidants such as catalase, SOD, and GSH. The decrease is possibly due to the protein oxidation caused by ethylparaben. Increased Peroxidase levels interact with enzymes and reduce enzyme activity, resulting in modifications in histidine residues and protein cross-link [58]. Similar findings were observed by Silva, Serrano [55] and Shah and Verma [54]. The results of oxidative stress and antioxidant enzymes indicate a reduction in the scavenging activity of antioxidant system on free radicals. Hence, ethylparaben causes oxidative stress in *L.rohita*.

DNA strand breakage, such as Olive tail moment (OTM) and percentage tail DNA to blood cells, was confirmed by comet assay in *L. rohita*. The results of the comet assay confirmed that ethyl paraben causes genotoxicity. Genotoxicity caused by parabens in human lymphocytes was previously evaluated by Güzel Bayülken, Ayaz Tüylü [59] which supports our results. Similarly genotoxicity caused by parabens was also evaluated by Martín, Peropadre [60] in the liver cells of monkeys.

## Conclusion

In fish, ethylparaben exposure over a longer period of time alters their hemato-biochemistry. As their concentration rises, so does their toxicity. Indicators of ethyl paraben's harmful effects, primarily on kidney, gills, and liver tissues, were also shown to have undergone histological modification. Ethylparaben causes oxidative stress, and, antioxidant enzyme activity is inhibited when exposed to ethylparaben. A comet assay used on blood cells exposed to ethylparaben also revealed genotoxicity. According to the study's findings, prolonged exposure to ethyl paraben harms aquatic life and changes its typical anatomical and physiological features, as well as oxidant and antioxidant enzyme activity and DNA damage.

## Supporting information

**S1 Data. Genotoxicity.**
(XLSX)

**S2 Data. Hematology and biochemical.**
(XLSX)

**S3 Data. Oxidative stress.**
(XLSX)

## Acknowledgments

The current study is a part of PhD research. The manuscript has not been published elsewhere.

## Author Contributions

**Conceptualization:** Hasnain Akmal, Muddasir Hassan Abbasi, Khurram Shahzad.

**Data curation:** Hasnain Akmal.

**Formal analysis:** Hasnain Akmal, Shabbir Ahmad.

**Investigation:** Khurram Shahzad.

**Methodology:** Hasnain Akmal, Shabbir Ahmad.

**Project administration:** Hasnain Akmal.

**Software:** Farhat Jabeen.

**Supervision:** Muddasir Hassan Abbasi, Khurram Shahzad.

**Visualization:** Farhat Jabeen.

**Writing – original draft:** Hasnain Akmal, Shabbir Ahmad.

**Writing – review & editing:** Muddasir Hassan Abbasi, Farhat Jabeen, Khurram Shahzad.

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
