## [Decision Letter · Decision Letter 0]

5 Nov 2023

PONE-D-23-29563A study on assessing the toxic effects of ethyl paraben on rohu (Labeo rohita)  using different biomarkers; Hemato-biochemical assays, Histology, Oxidant and Antioxidant activity and GenotoxicityPLOS ONE

Dear Dr. Shahzad,

Thank you for submitting your manuscript to PLOS ONE. After careful consideration, we feel that it has merit but does not fully meet PLOS ONE’s publication criteria as it currently stands. Therefore, we invite you to submit a revised version of the manuscript that addresses the points raised during the review process.

We look forward to receiving your revised manuscript.

Kind regards,

Amel Mohamed El Asely

Academic Editor

PLOS ONE

Journal Requirements:

2. Thank you for stating the following financial disclosure: NO

"The current study is part of my PhD research. No financial assistance was found during the study. There are no authors who have a conflict of interest. The manuscript has not been published elsewhere."

Please remove any funding-related text from the manuscript and let us know how you would like to update your Funding Statement. Currently, your Funding Statement reads as follows: NO

6. Please ensure that you refer to Figure 3 in your text as, if accepted, production will need this reference to link the reader to the figure.

7. Please include a caption for figure 7.

8. Please upload a copy of Figure 7, to which you refer in your text on page 19 in PDF submission. If the figure is no longer to be included as part of the submission please remove all reference to it within the text.

9. We note you have included a table to which you do not refer in the text of your manuscript. Please ensure that you refer to Table 2 in your text; if accepted, production will need this reference to link the reader to the Table.

Reviewers' comments:

Reviewer's Responses to Questions

**Comments to the Author**

1. Is the manuscript technically sound, and do the data support the conclusions?

Reviewer #1: Partly

Reviewer #2: Partly

2. Has the statistical analysis been performed appropriately and rigorously? 

Reviewer #1: Yes

Reviewer #2: Yes

3. Have the authors made all data underlying the findings in their manuscript fully available?

Reviewer #1: Yes

Reviewer #2: Yes

4. Is the manuscript presented in an intelligible fashion and written in standard English?

Reviewer #1: Yes

Reviewer #2: No

5. Review Comments to the Author

Reviewer #1: The manuscript entitled (A study on assessing the toxic effects of ethyl paraben on rohu (Labeo rohita) using different biomarkers; Hemato-biochemical assays, Histology, Oxidant and Antioxidant activity and Genotoxicity) needs major changes before being published

General comments:

The manuscript needs language editing

The introduction section should be improved

The methodology needs more details

The aim of work not clear

Specific comments

Abstract section

-line 18, insert a full stop after (Products)

-line 19 to 21 rephrase

-the abstract should be informative, the scientific name of fish should be added, number of fish used and their average weight, number of fish groups, number of replicates, and the experimental duration.

The methodology should mention briefly in the abstract.

-Add a space between T0(0.00 µg/L).

-the full name of all abbreviations written in the abstract should be written

- (Due to higher oxidative enzymes activity antioxidant enzymes potential to overcome their stress activity reduced) this sentence should be deleted, no explanations were written in the abstract

-which level of ethyl paraben that produced the results written in the abstract

Keywords: remove the word, aquatic toxicology

Introduction section

-the introduction section lack the information about the toxic impact of paraben on fish tissue, health

Also the residues of such paraben in water bodies, how paraben gain access to the water bodies. The uses of paraben in agriculture practices

-the significance of the studied fish (rohu, Labeo rohita) should be added in the introduction section

Why the authors used this fish as toxicological model

-the aim of work not clear because the problem not clarified well in the introduction section

-add the following references in the introduction section, they will be useful

1- Ahmed, S.A., Ibrahim, R.E., Elshopakey, G.E., Khamis, T., Abdel-Ghany, H.M., Abdelwarith, A.A., Younis, E.M., Davies, S.J., Elabd, H., Elhady, M.J.F., Immunology, S., 2022. Immune-antioxidant trait, growth, splenic cytokines expression, apoptosis, and histopathological alterations of Oreochromis niloticus exposed to sub-lethal copper toxicity and fed thyme and/or basil essential oils enriched diets. 131, 1006-1018.

2- Ibrahim, R.E., El-Houseiny, W., Behairy, A., Mansour, M.F., Abd-Elhakim, Y.M., 2019. Ameliorative effects of Moringa oleifera seeds and leaves on chlorpyrifos-induced growth retardation, immune suppression, oxidative stress, and DNA damage in Oreochromis niloticus. Aquaculture 505, 225-234.

3- Mohamed, W.A., El-Houseiny, W., Ibrahim, R.E., Abd-Elhakim, Y.M., 2019. Palliative effects of zinc sulfate against the immunosuppressive, hepato-and nephrotoxic impacts of nonylphenol in Nile tilapia (Oreochromis niloticus). Aquaculture 504, 227-238.

4- Ibrahim, R.E., Elbealy, M.A., Salem, G.A., Abdelwarith, A.A., Younis, E.M., Wagih, E., Elkady, A.A., Davies, S.J. and Rahman, A.N.A., 2023. Acute mancozeb-fungicide exposure induces neuro-ethology disruption, health disorders, and immune-oxidative dysfunction in Nile tilapia (Oreochromis niloticus). Aquatic Toxicology, 261, p.106630.

5- Ibrahim, Rowida E., Heba I. Ghamry, Saed Ayidh Althobaiti, Daklallah A. Almalki, Medhat S. Shakweer, Mona A. Hassan, Tarek Khamis, Heba M. Abdel-Ghany, and Shaimaa AA Ahmed. "Moringa oleifera and Azadirachta indica Leaves enriched diets mitigate chronic oxyfluorfen toxicity induced immunosuppression through disruption of pro/anti-inflammatory gene pathways, alteration of antioxidant gene expression, and histopathological Alteration in Oreochromis niloticus." Fishes 8, no. 1 (2022): 15.

Material and method section

1-line 91, replace made with assigned

2-number of fish and weight should be mentioned

3-what is the LC50 value for ethyl parapen, why authors evaluated it for 48 hours not for 96 hours

4- What is the relation between the lC50 and the different doses of parapen used in the experiment?

5-line 109, add the dose of clove oil and reference

6-the number of samples should be written for each group in all methodology (blood samples, histological samples, ….etc)

7-the procedures of histological examinations is lack many details, so superficial

The details should be added

8- All abbreviations should be defined for the first time

9- For the statistical analysis, the normality test should be performed before the ANOVA

Results

1- All tables titles should be rephrased and rewritten

2- Table footnotes should be added and the full names of all measured parameters should be inserted in the footnote

3- Line 196, the author written (propyl paraben), the authors used ethyl paraben or propyl paraben

4- All data in the figures and tables should be carefully revised by the authors

Discussion section

This part should be revised and improved

Line 320-321 why written in italic form

Many language errors are present

Conclusion section: this section not present

Reviewer #2: 1. The abstract has not been written in a clear and concise manner. For example the paragraph starting with ethyl paraben induced oxidative .....; is not stating the findings clearly.

2. There is reference to genotoxicity in the abstract. However, it is not clear which marker for genotoxicity was used to arrive at this conclusion. If this was based on the COMET Assay, then indicate that "based on the comet assay, ....

3. Under methods: it is not clear how the dosages of 2000, 4000, and 6000 were chosen. Is this based on actual estimated concentrations of paraben in contaminated waterways? How much paraben is fish exposed to in water? Do they bio-accumulate in fish tissues? What steps were taken to ensure that prior exposure to paraben and related chemicals did not occur in regard to fish used for the study? This needs to come out clearly.

4. Quite significantly, the manuscript has recurring grammar issues, and poor prose/sentence construction. For example, the sentence that reads..." all experiments were formed.... The word "formed" is quite out of place. This is repeated throughout (line 75). Experiments are "conducted or performed" etc... Do not use don't..., rather use do not etc...

5. Did the study investigate levels of glutathione or glutathione reductase? In the results, the sub-title is GSH, which is used to abbreviate glutathione and not glutathione reductase. In the discussion and figures, references have been made in regard to antioxidants including glutathione (GSH). Where does glutathione reductase come in line 140-144?

6. Hematology data has been presented in a table format. This is not correct. Dose response studies are best presented using line graphs to easily decipher the trend. This observation applies to antioxidant analysis (GSH, TBARS, ROS, SOD, CAT etc..)

7. The data on antioxidant assay is quite confusing for liver/kidney. ROS is going up, consistent with TBARS; but antioxidant enzymes that are meant to be up-regulated in response to exposure to chemical toxicants such as paraben are actually being down-regulated. This is a paradox. Note that there is a dose-dependent rise in ROS. Antioxidants are expected to rise in concert to the rise in ROS. For this study, antioxidants are going down as the dose of paraben goes up. However, many studies have demonstrated depletion of GSH levels in response to overwhelming chemical-induced oxidative stress. This may explain the decrease in GSH. Not sure about the SOD, CAT etc..,. This scenario may be possible when there is overwhelming oxidative stress, depleting all the antioxidants. Clarify this phenomenon. Notably, liver function markers are all up-regulated, signaling severe paraben-driven liver injury.

8. Data for hematology shows evidence for paraben-driven significant up-regulation of platelets. This elevation will have serious implications for blood-related disorders like DVT and patients who may be on blood thinning medications. This aspect needs to be discussed more broadly in the discussion section to highlight implications in various circumstances. Infact, RBCs and HGB will be expected to be down-regulated due to a possible paraben-driven impediment to hematopoiesis.

9. It is not clear why paraben will result in elevation of HGB, MCV, MCHC and platelets.

10. Notably, at low dosages, the effect of paraben on blood indices was low, validating the premise of this study. This, probably needs to come out well in the discussion.

11. It is very important that the grammar issues are corrected, and the manuscript is written carefully in a concise and logical manner using appropriate english.

12. There is need for a more concise title tied to the major findings of the study.

6. PLOS authors have the option to publish the peer review history of their article (what does this mean?). If published, this will include your full peer review and any attached files.

Reviewer #1: No

Reviewer #2: **Yes: **Alfred Orina Isaac

---

## [Author Response · Author response to Decision Letter 0]

20 Nov 2023

01 Please ensure that your manuscript meets PLOS ONE's style requirements, including those for file naming. Responce: Revised and corrected

02 Please clarify the sources of funding (financial or material support) for your study. List the grants or organizations that supported your study, including funding received from your institution, .... 

Responce: The authors received no specific funding for this work.”

03 Please remove any funding-related text from the manuscript. 

Responce: Revised, edited and corrected

04 In your Data Availability statement, you have not specified where the minimal data set underlying the results described in your manuscript can be found. PLOS defines a study's minimal data set as the underlying data used to reach the conclusions drawn in the manuscript and any additional data required to replicate the reported study findings in their entirety. 

Respone: All data set are present in submitted manuscript. Raw data from this study are available upon request.

05 We note that you have indicated that data from this study are available upon request. PLOS only allows data to be available upon request if there are legal or ethical restrictions on sharing data publicly. 

Responce: No legal or ethical restrictions on sharing data publicaly.

06 Please ensure that you refer to Figure 3 in your text as, if accepted, production will need this reference to link the reader to the figure. 

Responce: corrected

07 Please include a caption for figure 7. 

Responce: corrected

08 Please upload a copy of Figure 7, to which you refer in your text on page 19 in PDF submission. If the figure is no longer to be included as part of the submission please remove all reference to it within the text. 

Responce: Revised, edited and corrected

09 We note you have included a table to which you do not refer in the text of your manuscript. Please ensure that you refer to Table 2 in your text; if accepted, production will need this reference to link the reader to the Table. 

Responce: Revised, edited and corrected

Respond to comments of Reviewer #1 

01 The manuscript needs language editing.

The introduction section should be improved.

The methodology needs more details.

The aim of work not clear 

Responce: Revised and corrected

02 Abstract section

-line 18, insert a full stop after (Products)

-line 19 to 21 rephrase

-the abstract should be informative, the scientific name of fish should be added, number of fish used and their average weight, number of fish groups, number of replicates, and the experimental duration. 

Responce: Revised and corrected

03 The methodology should mention briefly in the abstract.

-Add a space between T0(0.00 µg/L). 

Responce: Revised and corrected

04 the full name of all abbreviations written in the abstract should be written

- (Due to higher oxidative enzymes activity antioxidant enzymes potential to overcome their stress activity reduced) this sentence should be deleted, no explanations were written in the abstract 

Responce: Revised and corrected

05 which level of ethyl paraben that produced the results written in the abstract 

06 Keywords: remove the word, aquatic toxicology 

Responce: Revised and corrected

07 Introduction section

-the introduction section lack the information about the toxic impact of paraben on fish tissue, health

Also the residues of such paraben in water bodies, how paraben gain access to the water bodies. The uses of paraben in agriculture practices

-the significance of the studied fish (rohu, Labeo rohita) should be added in the introduction section

Why the authors used this fish as toxicological model

-the aim of work not clear because the problem not clarified well in the introduction section 

Responce: Revised and corrected

08 Material and method section

1-line 91, replace made with assigned

2-number of fish and weight should be mentioned

3-what is the LC50 value for ethyl parapen, why authors evaluated it for 48 hours not for 96 hours

4- What is the relation between the lC50 and the different doses of parapen used in the experiment?

5-line 109, add the dose of clove oil and reference

6-the number of samples should be written for each group in all methodology (blood samples, histological samples, ….etc)

7-the procedures of histological examinations is lack many details, so superficial

The details should be added

8- All abbreviations should be defined for the first time

9- For the statistical analysis, the normality test should be performed before the ANOVA 

Responce: Revised and corrected

09 Results

1- All tables titles should be rephrased and rewritten

2- Table footnotes should be added and the full names of all measured parameters should be inserted in the footnote

3- Line 196, the author written (propyl paraben), the authors used ethyl paraben or propyl paraben

4- All data in the figures and tables should be carefully revised by the authors

Discussion section

This part should be revised and improved

Line 320-321 why written in italic form

Many language errors are present 

Responce: Revised and corrected

10 Conclusion section: this section not present 

Responce: Revised and corrected

Respond to comments of Reviewer #2 

01 The abstract has not been written in a clear and concise manner. For example the paragraph starting with ethyl paraben induced oxidative .....; is not stating the findings clearly. 

Responce: Revised and corrected

02 There is reference to genotoxicity in the abstract. However, it is not clear which marker for genotoxicity was used to arrive at this conclusion. If this was based on the COMET Assay, then indicate that "based on the comet assay, 

Responce:Revised and corrected

03 Under methods: it is not clear how the dosages of 2000, 4000, and 6000 were chosen. Is this based on actual estimated concentrations of paraben in contaminated waterways? How much paraben is fish exposed to in water? Do they bio-accumulate in fish tissues? What steps were taken to ensure that prior exposure to paraben and related chemicals did not occur in regard to fish used for the study? This needs to come out clearly. 

Responce: As per previous work, LC50 was determined. So, according to this, LC50 doses were divided into ½, 1/3, and ¼. These doses were actually based on laboratory work, and their estimation in natural or contaminated water is under process. We did not work on the estimation of paraben in fish. Fish were taken from fresh water fish form, where treated water was used.

04 Quite significantly, the manuscript has recurring grammar issues, and poor prose/sentence construction. For example, the sentence that reads..." all experiments were formed.... The word "formed" is quite out of place. This is repeated throughout (line 75). Experiments are "conducted or performed" etc... Do not use don't..., rather use do not etc... 

Responce:Revised and corrected

05 Did the study investigate levels of glutathione or glutathione reductase? In the results, the sub-title is GSH, which is used to abbreviate glutathione and not glutathione reductase. In the discussion and figures, references have been made in regard to antioxidants including glutathione (GSH). Where does glutathione reductase come in line 140-144? 

06 Hematology data has been presented in a table format. This is not correct. Dose response studies are best presented using line graphs to easily decipher the trend. This observation applies to antioxidant analysis (GSH, TBARS, ROS, SOD, CAT etc..) 

Responce: Revised and corrected

07 The data on antioxidant assay is quite confusing for liver/kidney. ROS is going up, consistent with TBARS; but antioxidant enzymes that are meant to be up-regulated in response to exposure to chemical toxicants such as paraben are actually being down-regulated. This is a paradox. Note that there is a dose-dependent rise in ROS. Antioxidants are expected to rise in concert to the rise in ROS. For this study, antioxidants are going down as the dose of paraben goes up. However, many studies have demonstrated depletion of GSH levels in response to overwhelming chemical-induced oxidative stress. This may explain the decrease in GSH. Not sure about the SOD, CAT etc..,. This scenario may be possible when there is overwhelming oxidative stress, depleting all the antioxidants. Clarify this phenomenon. Notably, liver function markers are all up-regulated, signaling severe paraben-driven liver injury. 

Responce: Explained in line 345-379

08 Data for hematology shows evidence for paraben-driven significant up-regulation of platelets. This elevation will have serious implications for blood-related disorders like DVT and patients who may be on blood thinning medications. This aspect needs to be discussed more broadly in the discussion section to highlight implications in various circumstances. Infact, RBCs and HGB will be expected to be down-regulated due to a possible paraben-driven impediment to hematopoiesis. 

Responce: Explained in line 285-290

09 It is not clear why paraben will result in elevation of HGB, MCV, MCHC and platelets. 

Responce: Revised and corrected

10 Notably, at low dosages, the effect of paraben on blood indices was low, validating the premise of this study. This, probably needs to come out well in the discussion. 

Responce: Revised and corrected

11 It is very important that the grammar issues are corrected, and the manuscript is written carefully in a concise and logical manner using appropriate english 

Responce: Revised and corrected

12 There is need for a more concise title tied to the major findings of the study. 

Responce: Revised and corrected

---

## [Decision Letter · Decision Letter 1]

8 Dec 2023

PONE-D-23-29563R1A study on assessing the toxic effects of ethyl paraben on rohu (Labeo rohita)  using different biomarkers; Hemato-biochemical assays, Histology, Oxidant and Antioxidant activity and GenotoxicityPLOS ONE

Dear Dr. Shahzad,

Thank you for submitting your manuscript to PLOS ONE. After careful consideration, we feel that it has merit but does not fully meet PLOS ONE’s publication criteria as it currently stands. Therefore, we invite you to submit a revised version of the manuscript that addresses the points raised during the review process.

We look forward to receiving your revised manuscript.

Kind regards,

Amel Mohamed El Asely

Academic Editor

PLOS ONE

Reviewers' comments:

Reviewer's Responses to Questions

**Comments to the Author**

1. If the authors have adequately addressed your comments raised in a previous round of review and you feel that this manuscript is now acceptable for publication, you may indicate that here to bypass the “Comments to the Author” section, enter your conflict of interest statement in the “Confidential to Editor” section, and submit your "Accept" recommendation.

Reviewer #1: All comments have been addressed

Reviewer #2: (No Response)

2. Is the manuscript technically sound, and do the data support the conclusions?

Reviewer #1: Yes

Reviewer #2: Partly

3. Has the statistical analysis been performed appropriately and rigorously? 

Reviewer #1: Yes

Reviewer #2: Yes

4. Have the authors made all data underlying the findings in their manuscript fully available?

Reviewer #1: Yes

Reviewer #2: Yes

5. Is the manuscript presented in an intelligible fashion and written in standard English?

Reviewer #1: Yes

Reviewer #2: No

6. Review Comments to the Author

Reviewer #1: the authors made all comments and the revised manuscript now accepted for publication

no further correction were needed

Reviewer #2: I gave detailed review proposals in my first analysis of this work. The authors have made reasonable effort to correct the figure for dose response studies as i guided. However, the grammar issues have not been addressed across the board starting with the abstract, through to the discussion section. A lot of work needs to be done here.

7. PLOS authors have the option to publish the peer review history of their article (what does this mean?). If published, this will include your full peer review and any attached files.

Reviewer #1: No

Reviewer #2: No

---

## [Editor Report · Decision Letter 2]

19 Feb 2024

PONE-D-23-29563R2A study on assessing the toxic effects of ethyl paraben on rohu (Labeo rohita)  using different biomarkers; Hemato-biochemical assays, Histology, Oxidant and Antioxidant activity and GenotoxicityPLOS ONE

Dear Dr. Shahzad,

Thank you for submitting your manuscript to PLOS ONE. After careful consideration, we feel that it has merit but does not fully meet PLOS ONE’s publication criteria as it currently stands. Therefore, we invite you to submit a revised version of the manuscript that addresses the points raised during the review process.

We look forward to receiving your revised manuscript.

Kind regards,

Amel Mohamed El Asely

Academic Editor

PLOS ONE
---

## [Author Response · Author response to Decision Letter 2]

23 Feb 2024

If applicable, we recommend that you deposit your laboratory protocols in protocols.io to enhance the reproducibility of your results.

answer: Not applicable

Answer: We have reviewed all references. No cited paper has been retracted.

While revising your submission, please upload your figure files to the Preflight Analysis and Conversion Engine (PACE) digital diagnostic tool,

Answer: Corrected and submitted in supporting information

---

## [Editor Report · Decision Letter 3]

2 Apr 2024

PONE-D-23-29563R3A study on assessing the toxic effects of ethyl paraben on rohu (Labeo rohita)  using different biomarkers; Hemato-biochemical assays, Histology, Oxidant and Antioxidant activity and GenotoxicityPLOS ONE

Dear Dr. Shahzad,

Thank you for submitting your manuscript to PLOS ONE. After careful consideration, we feel that it has merit but does not fully meet PLOS ONE’s publication criteria as it currently stands. I have observed that your response to the reviewers' comments was not addressed carefully, and the quality of English in your manuscript still needs improvement. Therefore, we recommend that you adhere to the instructions by providing detailed responses to the reviewers in a separate file and enhancing English in your manuscript. The attached file contains the comments from the reviewers. Please carefully consider every point and respond to them individually, addressing each one separately.

We look forward to receiving your revised manuscript.

Kind regards,

Amel Mohamed El Asely

Academic Editor

PLOS ONE
---

## [Author Response · Author response to Decision Letter 3]

4 Apr 2024

Respond to comment Response

01 I have observed that your response to the reviewers' comments was not addressed carefully, and the quality of English in your manuscript still needs improvement. 

Ans: Revised and improved English.

02 Please review your reference list to ensure that it is complete and correct. If you have cited papers that have been retracted, please include the rationale for doing so in the manuscript text, or remove these references and replace them with relevant current references.

 Ans: We cross checked all references and we haven’t found any retracted reference. Please let us know any retracted reference. 

03 While revising your submission, please upload your figure files to the Preflight Analysis and Conversion Engine (PACE) digital diagnostic tool. 

Ans: Figures have been uploaded and resized according to plos requirements and uploaded as figure and raw data in supporting information title “figure. Rar”.

---

## [Editor Report · Decision Letter 4]

10 Apr 2024

A study on assessing the toxic effects of ethyl paraben on rohu (Labeo rohita)  using different biomarkers; Hemato-biochemical assays, Histology, Oxidant and Antioxidant activity and Genotoxicity

PONE-D-23-29563R4

Dear Dr. Shahzad,

We’re pleased to inform you that your manuscript has been judged scientifically suitable for publication and will be formally accepted for publication once it meets all outstanding technical requirements.

Kind regards,

Amel Mohamed El Asely

Academic Editor

PLOS ONE